# Critical Thinking, Generalized Anxiety in Satisfaction with Studies: The Mediating Role of Academic Self-Efficacy in Medical Students

**DOI:** 10.3390/bs13080665

**Published:** 2023-08-09

**Authors:** Elbert Huamán-Tapia, Robinson B. Almanza-Cabe, Liset Z. Sairitupa-Sanchez, Sandra B. Morales-García, Oriana Rivera-Lozada, Alcides Flores-Paredes, Wilter C. Morales-García

**Affiliations:** 1Unidad de Educación, Escuela de Posgrado, Universidad Peruana Unión, Lima 15001, Peru; elberthuaman@upeu.edu.pe; 2Escuela Profesional Gestión Pública y Desarrollo Social, Universidad Nacional de Moquegua, Moquegua 18001, Peru; ralmanzac@unam.edu.pe; 3Escuela Profesional de Psicología, Facultad de Ciencias de la Salud, Universidad Peruana Unión, Lima 15001, Peru; lisetsairitupa@upeu.edu.pe; 4Facultad de Farmacia y Bioquímica, Universidad Científica del Sur, Lima 15067, Peru; 100097321@cientifica.edu.pe; 5South American Center for Education and Research in Public Health, Universidad Norbert Wiener, Lima 15046, Peru; oriana.rivera@uwiener.edu.pe; 6Escuela Profesional de Educación Física, Universidad Nacional del Altiplano Puno, Puno 21001, Peru; alcidesflores@unap.edu.pe; 7Escuela de Medicina Humana, Facultad de Ciencias de la Salud, Universidad Peruana Unión, Lima 15001, Peru; 8Escuela de Posgrado, Universidad Peruana Unión, Lima 15001, Peru; 9Facultad de Teología, Universidad Peruana Unión, Lima 15001, Peru

**Keywords:** critical thinking, anxiety, satisfaction, academic, self-efficacy

## Abstract

Background: The academic and emotional challenges faced by medical students can affect critical thinking and may also contribute to the development of increased generalized anxiety. Similarly, critical thinking and generalized anxiety can impact study satisfaction through the mediating mechanism of academic self-efficacy. Objective: The aim of this study was to assess the mediating role of academic self-efficacy between critical thinking and generalized anxiety in study satisfaction among medical students. Methods: A cross-sectional and explanatory study was conducted involving 259 Peruvian medical students aged between 18 and 35 (M = 20.29, SD = 2.84). The evaluation was based on self-reported questionnaires covering critical thinking, generalized anxiety, academic self-efficacy, and study satisfaction. Furthermore, a structural equation modeling (SEM) and mediation approach was employed to examine the relationships between variables. Results: The results showed an adequate fit of the model [χ^2^ (87) = 155, *p* < 0.001, CFI = 0.93, TLI = 0.92, RMSEA = 0.05 (CI: 0.04–0.07), SRMR = 0.07], demonstrating the impact of critical thinking and generalized anxiety. It was confirmed that academic self-efficacy has a positive effect on study satisfaction. Moreover, the mediating role of academic self-efficacy was confirmed between critical thinking and study satisfaction, as well as between generalized anxiety and study satisfaction. Conclusions: Due to the high academic load on medical students, academic self-efficacy plays a mediating role in the relationship between critical thinking, generalized anxiety, and satisfaction with studies. The development of educational strategies will help to promote critical thinking and academic self-efficacy, as well as provide support to students with generalized anxiety, to enhance study satisfaction.

## 1. Introduction

Medical education is a challenging and rigorous field that requires the acquisition of advanced cognitive skills, such as critical thinking to ensure the preparation of competent and capable healthcare professionals. Critical thinking, characterized as a reflective and reasonable process that focuses on making decisions based on rigorous evaluation of evidence [1], provides the necessary tools to address and resolve professional dilemmas, enhancing the ability to manage and tolerate medical uncertainty [2,3]. It not only improves intuitive reasoning and experiential knowledge accumulation but also emerges as an essential competency in the academic domain for making decisions in complex situations, translating into benefits in various aspects of students’ lives and future careers [4,5,6].

However, medical students have higher rates of anxiety compared to the general population and their peers due to the academic and emotional demands of medical programs [7,8]. This chronic disorder, characterized by heightened sensitivity to stress and excessive and persistent worry [9,10], may be associated with poor academic performance and have a negative impact on students’ well-being and mental health [11,12]. In this regard, high levels of anxiety have a negative relationship with academic performance, as students react to low grades with feelings of failure and low self-esteem in environments with greater academic demands [13].

Furthermore, the well-being of medical students is affected by medical training, and their quality of life decreases during education [14]. Satisfaction with studies has been shown to be a crucial factor in academic success [15], and it can be influenced by both the ability to think critically and levels of generalized anxiety [16,17]. Within this framework, academic self-efficacy can play an essential, mediating role in the relationship between critical thinking, generalized anxiety, and satisfaction with studies. Academic self-efficacy has been linked to better academic performance, higher self-efficacy, and satisfaction with studies in university students [15,18,19,20,21].

Despite growing evidence of the relevance of critical thinking and generalized anxiety in educational contexts, there is a gap in the literature regarding the interaction of these variables in the specific context of medical education. Exploring these relationships is justified due to their potential to improve medical education and the well-being of medical students, as well as to develop more effective interventions that support students along this path. Additionally, although academic self-efficacy has been extensively studied in general educational contexts [22,23], its role as a mediator between critical thinking, generalized anxiety, and satisfaction with studies in medical students has not been adequately explored. Therefore, this study aims to fill that gap in the existing literature.

## 2. Literature Review

### 2.1. Critical Thinking

Critical thinking is a metacognitive process that encompasses a variety of underlying skills and is essential in both educational and social contexts. It is considered a key skill for informed decision-making, problem-solving, and forming logical conclusions [1,2]. It allows for the development of a particularly critical ability in disciplines such as medicine, where professionals often need to evaluate and synthesize complex data to gain a deeper understanding of information [3,4]. Problem-based learning in medical education has been identified as an effective strategy for fostering the development of critical thinking and transferable skills [3]. This pedagogical approach motivates students to construct their own knowledge and take responsibility for their learning through an active and self-directed focus [4]. Furthermore, various factors, such as the learning environment and teacher support, can influence the development of critical thinking [5]. Thus, active participation and collaboration in the classroom fosters knowledge construction and critical reflection, promoting high-level cognitive skills such as analysis, synthesis, and evaluation, as well as argumentation and the use of evidence to justify hypotheses [6]. Moreover, there is a clear and consistent relationship between achievement goals, self-efficacy, study strategies, and critical thinking in medical students, where mastery goals and a deep approach to information processing are positively related to critical thinking and academic performance [7].

### 2.2. Generalized Anxiety

Anxiety is a mental health disorder characterized by fear, excessive worry, and a constant feeling of being overwhelmed, affecting people’s daily lives [24]. Medical students may experience persistent worries about their academic performance, which can contribute to the onset of generalized anxiety and other psychological morbidities [25]. Generalized anxiety has a high prevalence among medical students, with studies reporting prevalence rates ranging from 7.7% to 65.5%, indicating that these students are at a higher risk of experiencing anxiety compared to their non-medical peers [26]. Thus, anxiety can be related to various stressors, a lack of balance, relationship difficulties, uncertainty about the future, and financial burdens [27,28]. In addition, generalized anxiety can have negative effects on academic performance and satisfaction with studies as it increases sensory perception processing, affects the balance between stimulus-driven and goal-directed behaviors, impairs inhibitory control, affects short and long-term memory, and can influence executive processes and decision-making [29]. In addition, generalized anxiety can negatively affect the emotional and physical well-being of medical students, increasing levels of depression and leading to a decrease in quality of life, as well as a higher prevalence of emotional exhaustion [30].

### 2.3. Satisfaction with Studies

Satisfaction with studies is a critical measure that refers to the subjective evaluation students make regarding the quality of the educational services they receive and to what extent these meet or exceed their expectations [8]. This indicator, essential for the success of educational institutions, can reflect both the quality of the curriculum and faculty, as well as the effectiveness of student support policies. Thus, it is crucial not only for higher education institutions but also for individual students, as their level of satisfaction can directly impact their engagement, performance, and retention [9]. Satisfaction with studies can also be influenced by factors such as academic difficulties, social adjustment, external commitments, and a sense of belonging [10].

In medical education, satisfaction with studies is a relevant factor due to the high prevalence of mental health problems among students. Distress, academic stress, ongoing difficulties balancing academic and personal responsibilities, and a lack of time for recreational and leisure activities can affect students’ quality of life and satisfaction with their studies [11,12]. In this regard, the relationship between these factors and academic performance can be complex and closely interconnected. Understanding these factors can be crucial to ensuring student engagement and success [11]. Understanding these connections can lead to the creation of more effective interventions that address the challenges faced by medical students and promote their overall well-being [13,14].

### 2.4. Academic Self-Efficacy

Academic self-efficacy refers to the beliefs one holds about their ability to organize and execute the actions necessary in handling academic situations. These beliefs influence the choices, effort, perseverance, and resilience of individuals in relation to academic tasks [31]. Self-efficacy and critical thinking are positively related, as students possessing well-developed critical thinking skills tend to present higher academic self-efficacy [19]. Additionally, it has been inferred that academic self-efficacy can act as a mediator between critical thinking and study satisfaction in university students [32,33]. However, more research is needed to specifically explore this mechanism in the population of medical students. In addition, generalized anxiety can negatively affect academic self-efficacy, as self-efficacy beliefs determine how people feel, think, are motivated, and behave, and anxiety can generate doubts about one’s capabilities and limit participation in challenging academic tasks [34]. Academic self-efficacy may play a mediating role in the relationship between generalized anxiety and satisfaction with studies [35]. Nevertheless, more research is needed to specifically examine this mechanism in the population of medical students and how it could be applied in interventions to improve satisfaction with studies in this population.

Based on our review of the literature, we explore the following research questions and hypotheses (Figure 1):

What is the relationship between critical thinking and academic self-efficacy in medical students? 

**Hypothesis 1.** 
*We expect a positive relationship between critical thinking and academic self-efficacy.*


What is the relationship between generalized anxiety and academic self-efficacy in medical students? 

**Hypothesis 2.** 
*Additionally, we anticipate a negative relationship between generalized anxiety and academic self-efficacy.*


What is the relationship between academic self-efficacy and satisfaction with studies among medical students? 

**Hypothesis 3.** 
*We also expect a positive relationship between academic self-efficacy and satisfaction with studies.*


Is the relationship between critical thinking and satisfaction with studies mediated by the academic self-efficacy of medical students? 

**Hypothesis 4a.** 
*We hypothesize that academic self-efficacy mediates the relationship between critical thinking and satisfaction with studies.*


Is the relationship between generalized anxiety and satisfaction with studies mediated by the academic self-efficacy of medical students? 

**Hypothesis 4b.** 
*We expect that academic self-efficacy mediates the relationship between generalized anxiety and satisfaction with studies.*


## 3. Methods

### 3.1. Design and Participants

An explanatory study was conducted, as it sought to explore the relationships between variables and understand how they relate to each other. Moreover, mediation effects in these studies can be assessed through a structural equation modeling (SEM) system [36]. Sample selection was conducted using a non-probabilistic sampling process. The sample size was determined using the effect size calculation in the Soper electronic tool [37]. This calculator takes into account the number of observed and latent variables in the structural equation model (SEM), the desired statistical significance level (α = 0.05), the anticipated effect size (λ = 0.3), and the level of statistical power (1 - β = 0.80). The calculation determined that the minimum required sample would be 137 participants. A total of 259 medical students participated, with ages ranging between 18 and 35 years (M = 20.29, SD = 2.84). Most were female, from the coast, and in their second study cycle (Table 1).

### 3.2. Procedure

This study was conducted in strict compliance with current ethical and research standards. The research protocol was meticulously reviewed and subsequently approved by the Ethics Committee of a Peruvian university, with the approval reference number CE-DGI-0061. To obtain the study sample, careful coordination was conducted with the responsible authorities in several universities in Peru. This procedure was carried out to ensure that all potential participants had full knowledge of this study and its purpose. Subsequently, informed consent forms were provided to the students through various digital means, including emails, WhatsApp groups, and Google Forms, thereby expanding the possibility of participation and ensuring that participants were fully informed. The established protocol ensured that participants fully understood that their participation in this study was completely voluntary and that they could withdraw at any time without any negative consequences. This premise of respecting the autonomy of the participant is a standard practice in ethical research. Lastly, at every stage of this study, the ethical principles outlined in the Helsinki Declaration were faithfully followed, including ensuring confidentiality and protecting participants’ personal data. This involved keeping all collected information anonymous and using it exclusively for the purposes of this research study.

### 3.3. Variables

Critical Thinking. The Spanish version of the Individual Generic Skills Questionnaire (CCGI) was used, which measures critical thinking skills [38]. The questionnaire consists of 10 items and uses a 5-point Likert scale, ranging from “strongly agree” to “strongly disagree”. The reliability was 0.739 according to Cronbach’s Alpha coefficient. This study presented adequate validity indices: χ^2^ = 66.870, df = 34, *p* = < 0.001, CFI = 0.93, TLI = 0.90, RMSEA = 0.06 and SRMR = 0.05.

Generalized Anxiety. The Generalized Anxiety Disorder Scale-2 (GAD-2) was used, which has two items [39] and uses a 4-point Likert response scale: Never = 0, several days = 1, more than half of the days = 2, and almost every day = 3. A version adapted to Peruvian Spanish was used, with a reliability for internal consistency value of α = 0.738 (95% CI, 0.699, 0.773) [40].

Academic Self-Efficacy. The Single Item Academic Self-Efficacy Scale (IUAA) was used. This item assesses academic self-efficacy, that is, the confidence the student has in their ability to efficiently carry out the academic tasks demanded by their academic life. The item is rated on a 5-point scale ranging from “Not at all confident” to “Very confident”. A version adapted to Peruvian Spanish was used, with the IUAA’s internal consistency reliability estimated at 0.86 [41].

Study Satisfaction Scale: The Peruvian Spanish-adapted version of the Study Satisfaction Scale (EBSE) was used. It consists of three items that assess the student’s satisfaction with their study habits, their performance, and their overall experience with their studies. The items are rated on a 5-point response scale ranging from “strongly disagree” to “strongly agree”. The scale’s consistency showed an adequate magnitude, with an alpha coefficient of 0.78 [40].

### 3.4. Statistical Analysis

The theoretical model was analyzed using the method of structural equation modeling with a multiple linear regression (MLR) estimator and is robust against deviations from inferential normality [42]. The model fit was evaluated using several indicators that are widely accepted in the field of applied statistics and social sciences. These indices include the Comparative Fit Index (CFI), Tucker–Lewis Index (TLI), Root Mean Square Error of Approximation (RMSEA), and Standardized Root Mean Square Residual (SRMR). CFI and TLI values above 0.90 are considered acceptable, indicating a good fit between the model and data [43]. Additionally, RMSEA values below 0.080 are considered indicative of a reasonable fit [44]. Lastly, SRMR values below 0.080 are considered adequate [45]. To evaluate reliability, we used Cronbach’s alpha coefficient (α) [46]. The computational tools used were “R” software version 4.1.2 and the “lavaan” library in its 06-10 version [47].

Regarding the evaluation of mediation, we used the “psych” package [48]. According to established guidelines, the M variable acts as a mediator between the independent X variable and the dependent Y variable. In this scheme, if M is causally between X and Y, it means that the M variable is influenced by X and, in turn, affects Y [15,16]. This implies that the indirect effect of X on Y occurs through M. This configuration allows us to construct a causal chain to evaluate the mediating effect of the M variable. To test this indirect effect, we applied a bootstrapping technique with 500 iterations.

## 4. Results

### 4.1. Preliminary Analysis

Table 2 presents the descriptive statistics for all variables, where critical thinking has the highest mean (31.97) with a standard deviation of 4.89. Additionally, the bivariate correlation matrix is shown, where study satisfaction is positively related to academic self-efficacy (r = 0.33, *p* < 0.01), critical thinking (r = 0.32, *p* < 0.01), and negatively with generalized anxiety (r = −0.14, *p* < 0.05). Similarly, academic self-efficacy is positively related to critical thinking (r = −0.36, *p* < 0.01) and negatively to generalized anxiety (r = −0.26, *p* < 0.01). Moreover, the relationship between critical thinking and generalized anxiety was not significant. In addition, it is observed that Cronbach’s alpha internal consistencies were found between values that range between 0.80 and 0.86.

### 4.2. Theoretical Model Analysis

In the analysis of the theoretical model (Figure 2), an adequate fit was obtained, χ^2^ (87) = 155, *p* < 0.001, CFI = 0.93, TLI = 0.92, RMSEA = 0.05 (CI: 0.04–0.07), SRMR = 0.07. Critical thinking was positively related to academic self-efficacy (β = 0.43, *p* < 0.01); thus, these results agree with Hypothesis 1. Similarly, generalized anxiety had a negative relationship with academic self-efficacy (β = −0.31, *p* < 0.01), thereby confirming Hypothesis 2. Hypothesis 3 was also confirmed, agreeing with the positive relationship between academic self-efficacy and satisfaction with studies (β = 0.35, *p* < 0.001).

### 4.3. Mediation Model

For the mediation analysis, bootstrapping of 5000 iterations was used (Table 3). As expected, the results confirmed Hypothesis 4a, in which academic self-efficacy mediated the relationship between critical thinking and satisfaction with studies (β = 0.19, *p* < 0.01). Similarly, Hypothesis 4b was confirmed, where self-efficacy mediated the relationship between generalized anxiety and satisfaction with studies (β = −0.11, *p* < 0.01).

## 5. Discussion

In the landscape of higher education, students encounter a variety of challenges that can be closely linked to deficiencies in key areas such as knowledge, attitude, and behavior. Dealing with individual causes alone is not sufficient; careful attention must also be paid to crucial elements of instructional design and the educational environment, as they represent fundamental aspects that can shape students’ academic experience [49]. In this intricate scenario, teaching strategies emerge as powerful allies. Techniques such as case studies, student collaboration, and fostering critical reflection become valuable tools that enable the development of critical thinking skills. This capacity not only promotes more assertive decision-making but also acts as a shield to avoid common errors in the learning process [50]. However, academic life is not exempt from emotional obstacles. Students may experience significant levels of anxiety, often driven by fear of failure and negative experiences associated with academic intensity. This emotional storm can exert a detrimental influence on their academic performance. In addition, external factors such as stress and lack of motivation can amplify these effects, resulting in lower performance [49]. In this regard, generalized anxiety can have a negative impact on satisfaction with studies by interfering with concentration, decision-making, and engagement in academic activities. However, when students have high academic self-efficacy, they are more likely to effectively use critical thinking and cope with increased anxiety, which in turn can enhance their satisfaction with their studies.

The findings of this study, focusing on the relationships of the research model, confirmed the first hypothesis, which demonstrates a positive relationship between critical thinking and academic self-efficacy in medical students. This finding is consistent with previous research that has highlighted the importance of critical thinking in shaping self-efficacy beliefs in academic contexts [51,52]. This dynamic interplay of critical thinking and self-efficacy constructs a robust scaffold for effective academic learning and further drives the acquisition of deep knowledge [51]. Developing critical thinking enhances students’ ability to tackle and resolve complex problems, a process that, in turn, strengthens confidence in their academic skills [53]. In the context of medical education, this relationship translates into tangible benefits for students. Those who develop critical thinking skills not only see improvements in their academic performance but also exhibit greater resilience in the face of challenging tasks. Rather than avoiding difficult situations, they feel more equipped to confront and overcome them [54]. Furthermore, adopting these skills promotes the application of higher-level learning strategies to their studies, further fueling their confidence and academic growth [55].

Additionally, the second hypothesis regarding the negative relationship between generalized anxiety and academic self-efficacy was confirmed. This result is consistent with previous research that has shown anxiety can undermine academic self-efficacy, negatively affecting academic performance [56,57]. In this regard, academic difficulties, such as imbalances between work, family, and academic responsibilities, as well as a lack of effective time and stress management strategies, are aspects that play a prominent role in students’ anxiety [35]. Anxiety can negatively impact active participation in learning and studying situations. This negative impact can lead to avoidance behaviors, isolation, and, ultimately, loneliness [58].

Similarly, the third hypothesis was confirmed, showing a positive relationship between academic self-efficacy and satisfaction with studies in medical students. This finding is consistent with previous studies that have indicated that academic self-efficacy is an important predictor of satisfaction with studies in various educational contexts [32,59]. This is because generalized anxiety in self-efficacy occurs through cognitive processing and interpretation of challenging experiences, verbal persuasions, and physiological reactions to stressful situations. Thus, students with higher academic self-efficacy are more confident in their abilities to face academic and clinical challenges, which in turn allows them to enjoy and find more satisfaction in their studies [60,61]. In addition, academic self-efficacy can influence students’ motivation, effort, and persistence, factors that are related to satisfaction with studies and overall academic success [62].

Finally, the results of this study confirmed the mediatory role of academic self-efficacy in the relationship between critical thinking and satisfaction with studies in medical students. This finding is consistent with similar previous research that has shown that confidence in one’s academic abilities influences how students use critical thinking and, in turn, affects their satisfaction with the learning process [51]. This is due to the confidence that students have in their developed academic skills, represented by self-efficacy, which impacts the way they develop and utilize critical thinking. This, in turn, can affect satisfaction with the learning process [63]. In addition, academic self-efficacy mediates the relationship between generalized anxiety and satisfaction with studies. Similar studies suggest that academic self-efficacy can influence positive emotions and the use of metacognitive learning strategies, which in turn impact academic performance. Thus, one could infer that academic self-efficacy acts as a mediator in the relationship between generalized anxiety and satisfaction with studies [64]. Moreover, students with higher academic self-efficacy tend to deal with anxiety more effectively, which enables them to maintain a high level of satisfaction with their studies, as academic self-efficacy can serve as a coping resource that helps students handle the stress and anxiety associated with medical education. Therefore, academic self-efficacy could play a crucial role in emotional regulation and enable more effective management of academic stress in medical students, highlighting the need to incorporate strategies that strengthen self-efficacy in educational programs.

### 5.1. Limitations

Some limitations were found in this study, such as the cross-sectional nature of these data. This indicates the need for future research to examine these relationships longitudinally to determine causality between variables. In addition, it is crucial to consider the diversity of educational and cultural contexts in which medical education takes place, which may influence the generalization of the results.

### 5.2. Implications and Future Research

Our findings provide additional support for Bandura’s social learning theory by highlighting the importance of self-efficacy and expanding the understanding of the mediation of self-efficacy between critical thinking, generalized anxiety, and satisfaction with studies. Rather than simply considering it as a result of learning experiences, our study suggests that self-efficacy can be influenced by the student’s ability to think critically and manage anxiety. Additionally, we recommend building theoretical models that demonstrate interacting constructs in different educational and cultural contexts to determine the generalization of these findings. Hence, future research should explore how academic self-efficacy develops and can be strengthened over time. It would also be useful to further investigate the relationships between critical thinking, academic self-efficacy, anxiety, and satisfaction with studies.

In addition, institution administrators can encourage a learning environment that stimulates critical thinking through pedagogical approaches that allow the development of analysis, problem-solving, and problem-based learning. They should also develop or strengthen counseling and wellness services for managing anxiety and improving self-efficacy. Teachers could be provided with training for greater understanding, implementation, and promotion of self-efficacy. Additionally, a system of timely and constructive feedback by teachers should be implemented for greater opportunities for self-assessment.

Furthermore, practical implications for medical education and well-being are provided. In this sense, educational administrators can incorporate various teaching strategies that promote critical thinking, such as problem-solving, case analysis, project-based learning, and group discussions. Promotion of psychological support in high-stress environments is also necessary, which allows for periodic mental health assessment and provides support to address this issue. The implementation of interventions for improving academic self-efficacy, developing coping skills, and self-regulation should also be considered. These can be useful for students who have a wide academic demand.

Future research could examine the efficacy of interventions that foster critical thinking and academic self-efficacy in medical students and assess their impact on satisfaction with studies and emotional well-being. In addition, it would be relevant to explore individual and cultural differences that may affect satisfaction with studies, to adapt interventions to the specific needs of students from various educational and cultural contexts. Furthermore, research studies could investigate how institutional factors, academic policies, teaching practices, school climate, and support from the university community and teachers can affect academic self-efficacy, anxiety, and satisfaction with studies.

## 6. Conclusions

The mediating role of academic self-efficacy in the relationship between critical thinking, generalized anxiety, and satisfaction with studies in medical students can inform educational and support strategies aimed at improving emotional well-being and academic satisfaction. This knowledge allows educators and medical program administrators to design interventions and educational strategies that promote the development of critical thinking, reduce anxiety, and strengthen academic self-efficacy, ultimately aiming to improve study satisfaction and academic success among medical students.

## Figures and Tables

**Figure 1 behavsci-13-00665-f001:**
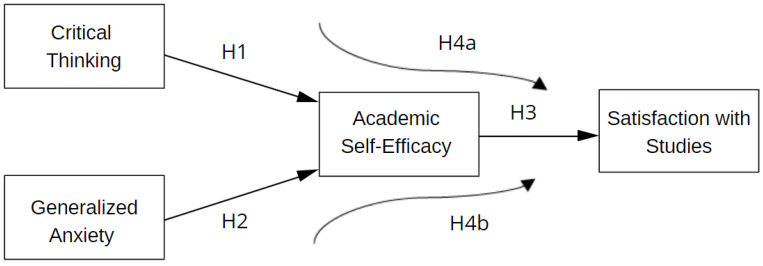
Theoretical model.

**Figure 2 behavsci-13-00665-f002:**
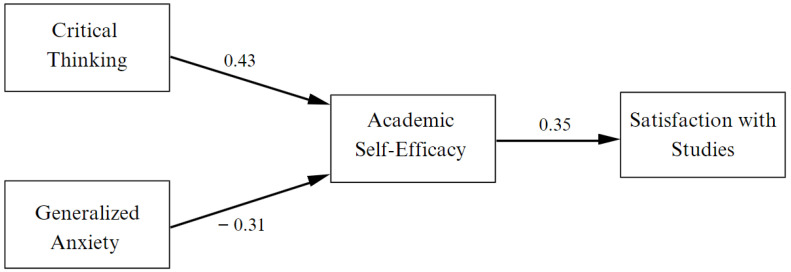
Results of the explanatory structural model.

**Table 1 behavsci-13-00665-t001:** Sociodemographic Characteristics.

Characteristics	n	%
Sex	Male	103
	Female	156
Origin	Coast	146
	Highlands	84
	Jungle	29
Study Cycle	1	8
	2	132
	3	1
	4	39
	5	3
	6	7
	8	5
	9	11
	10	53

**Table 2 behavsci-13-00665-t002:** Descriptive statistics, reliability, and correlations for study variables.

Variables	M	SD	α	1	2	3	4
1. Satisfaction with Studies	9.07	2.53	0.86	-			
2. Academic Self-efficacy	3.67	0.93	-	0.33 **	-		
3. Critical Thinking	31.97	4.89	0.81	0.32 **	0.36 **	-	
4. Generalized Anxiety	2.95	1.53	0.80	−0.14 *	−0.26 **	0.07	-

Note: M = Mean, SD = Standard deviation, α = Alpha coefficient, All correlations are statistically significant (* *p* < 0.05., ** *p* < 0.01.).

**Table 3 behavsci-13-00665-t003:** Research hypotheses on indirect effects and their estimates.

				95%CI	
Hypothesis	Path in the Mode	β	*p*	LL	UL
Hypothesis 4a	Critical Thinking → Academic Self-Efficacy → Study Satisfaction	0.19	<0.01	0.07	0.15
Hypothesis 4b	Generalized Anxiety → Academic Self-Efficacy → Study Satisfaction	−0.11	<0.01	−0.04	−0.10

## Data Availability

Data are available upon request from the author by correspondence.

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
