# Peer review of "Critical Thinking, Generalized Anxiety in Satisfaction with Studies: The Mediating Role of Academic Self-Efficacy in Medical Students"

_behavsci, 2023, doi:10.3390/bs13080665_

Round 1
Reviewer 1 Report
Dear Author,
The researcher stated that the academic and emotional challenges faced by medical students can affect critical thinking, and may also contribute to the development of increased generalized anxiety. Like- wise, critical thinking and generalized anxiety can impact study satisfaction through the mediating mechanism of academic self-efficacy. Objective: The aim of this study was to assess the mediating role of academic self-efficacy between critical thinking and generalized anxiety in study satisfaction among medical students. Methods: A cross-sectional and explanatory study was conducted involving 259 Peruvian medical students aged between 18 and 35 years (M=20.29, SD=2.84). Evaluation was based on self-reported questionnaires covering critical thinking, generalized anxiety, academic self-efficacy, and study satisfaction. Furthermore, a structural equation modeling (SEM) and mediation approach was employed to examine the relationships between variables. Results: The results showed an adequate fit of the model [x2 (87) = 155, p < .001, CFI = 0.93, TLI=0.92, RMSEA = 0.05 (CI: 0.04- 0.07), SRMR = 0.07], demonstrating the impact of critical thinking and generalized anxiety. It was confirmed that academic self-efficacy has a positive effect on study satisfaction. Moreover, the mediating role of academic self-efficacy was confirmed between critical thinking and study satisfaction, as well as between generalized anxiety and study satisfaction. Conclusions: Due to the high academic load on medical students, academic self-efficacy plays a mediating role in the relationship between critical thinking, generalized anxiety, and satisfaction with studies. The development of educational strategies will help to promote critical thinking and academic self-efficacy, as well as provide support to students with generalized anxiety, to enhance study satisfaction. Indeed, the current research paper offers a good insight related to the studied research along with the research objectives, questions and hypotheses. The research paper demonstrates an adequate understanding of the relevant literature in the field and cites an appropriate range of literature sources. Methodology is not clear; specifically in the population and sampling methods. 137 participants are not enough to generalize the results. Therefore, more data should be collected. However, analyses and findings are presented in a very good manner as to present new ideas. Also, the research has not rigor theoretical and managerial implications. Thus, the paper needs improvements in order to meet the standards of the journal.
Author Response
Reviewer 1
Methodology is not clear; specifically in the population and sampling methods. 137 participants are not enough to generalize the results. Therefore, more data should be collected.
A. Thank you for your comments, we have clarified what was indicated, the 137 indicated the estimate of the sample that was made using the Soper electronic calculator. However, a total of 259 students participated in the study.
“The sample selection was carried out through a non-probabilistic sampling process. The sample was determined through the calculation of effect size using the Soper electronic tool (Soper, 2022). This calculator considers the number of observed and latent variables in the structural equation model (SEM), the anticipated effect size (λ = 0.3), the desired statis-tical significance (α = 0.05) and the level of statistical power (1 - β = 0.80), the calculation determined that the minimum required sample would be 137 participants. A total of 259 medical students participated, with ages ranging between 18 and 35 years (M=20.29, SD= 2.84”
However, analyses and findings are presented in a very good manner as to present new ideas. Also, the research has not rigor theoretical and managerial implications. Thus, the paper needs improvements in order to meet the standards of the journal.
A. Thanks for your feedback, changes were made throughout the section.

Reviewer 2 Report
Research design in this article lacks clarity. The researchers need to more clearly define what they did in order for the reader to understand. Revisions to improve clarity of study procedures appear warranted. Moreover, present research questions and their associated hypotheses together.
Please clearly connect analyses to the research questions and/or hypotheses they seek to answer.
Additionally, provide details in relation with participants. Please include additional demographic information related to socio-economic status, race-ethnicity, etc. if available. Additional explanation and description should be provided for each of the measures included in the study. Please include psychometric evidence supporting use of these measures if available. If unavailable, please provide a rationale for their use.
Please reorganize content into “Results” and “Discussion” sections. In each section, clearly connect results and discussion content to the research question or hypothesis they address.
Implications needed to be clearly presented and discussed in relation with the opted study derived results.
Please review the article for grammar/syntax errors. Please review the article for APA errors. APA formatting rules (e.g., citations, use of parentheses, manuscript sections/headings/subheadings). Please conform for more conventional APA recommended headings and subheadings (i.e., introduction, Method [Participants, Measures, Procedures, Analysis Plan], Results, Discussion, Limitations and Future Directions, Conclusions).
Minor editing of English language required
Author Response
Reviewer 2
Research design in this article lacks clarity. The researchers need to more clearly define what they did in order for the reader to understand. Revisions to improve clarity of study procedures appear warranted. Moreover, present research questions and their associated hypotheses together.
A. Thank you for your suggestion changes were made to the text:
“Based on our review of the literature, we explore the following research questions and hypotheses:
What is the relationship between critical thinking and academic self-efficacy in medical students? We expect a positive relationship between critical thinking and academic self-efficacy (Hypothesis 1).
What is the relationship between generalized anxiety and academic self-efficacy in medical students? Additionally, we anticipate a negative relationship between generalized anxiety and academic self-efficacy (Hypothesis 2).
What is the relationship between academic self-efficacy and satisfaction with studies among medical students? We also expect a positive relationship between academic self-efficacy and satisfaction with studies (Hypothesis 3).
Is the relationship between critical thinking and satisfaction with studies mediated by the academic self-efficacy of medical students? We hypothesize that academic self-efficacy mediates the relationship between critical thinking and satisfaction with studies (Hypothesis 4a).
Is the relationship between generalized anxiety and satisfaction with studies mediated by the academic self-efficacy of medical students? We expect that academic self-efficacy mediates the relationship between generalized anxiety and satisfaction with studies (Hypothesis 4b).”
Please clearly connect analyses to the research questions and/or hypotheses they seek to answer.
A. Thank you for your suggestion changes were made to the text:
Additionally, provide details in relation with participants. Please include additional demographic information related to socio-economic status, race-ethnicity, etc. if available. Additional explanation and description should be provided for each of the measures included in the study. Please include psychometric evidence supporting use of these measures if available. If unavailable, please provide a rationale for their use.
A. Thank you for your comment. Regarding socioeconomic status, it was not considered because in Peru, medical students are predominantly from the upper-middle class, so it was not the focus of this study. Race or ethnic origin was also not considered because it is not a significant factor in Peru, but the region of origin was considered.
In terms of psychometric evidence, all measures except critical thinking have been validated in Peru and specifically in the context of university students. In this sense, it was considered appropriate to add validity indices to the Critical Thinking scale, and it was specified that the instruments have already been reported as valid in the Peruvian context.
Please reorganize content into “Results” and “Discussion” sections. In each section, clearly connect results and discussion content to the research question or hypothesis they address.
A. Corrections were made in the text
Implications needed to be clearly presented and discussed in relation with the opted study derived results.
A. The implications section was organized
Please review the article for grammar/syntax errors. Please review the article for APA errors. APA formatting rules (e.g., citations, use of parentheses, manuscript sections/headings/subheadings). Please conform for more conventional APA recommended headings and subheadings (i.e., introduction, Method [Participants, Measures, Procedures, Analysis Plan], Results, Discussion, Limitations and Future Directions, Conclusions).
A. Thank you for your suggestion formatting corrections were made throughout the text

Reviewer 3 Report
- Description of hypothesis H4b doesn't seem to be accurate when compared to Figure 1.
- Description of results (4.1. Preliminary Analysis) does not match with Table 2. Based on table 2, r = -0.26 is a correlation coefficient between generalized anxiety (4) and academic self-efficacy (2), not critical thinking. Based on the smame table, generalized anxiety and critical thinking are not correlated r = 0.07 with no statistical significance.
- Table can be presented a bit better. What M and DE are, should be explained in the descriptive statistics.
- Please specify what MLR stands for.
Author Response
- Description of hypothesis H4b doesn't seem to be accurate when compared to Figure 1.
A. Thanks for the suggestions, changes were made to the text
- Description of results (4.1. Preliminary Analysis) does not match with Table 2. Based on table 2, r = -0.26 is a correlation coefficient between generalized anxiety (4) and academic self-efficacy (2), not critical thinking. Based on the smame table, generalized anxiety and critical thinking are not correlated r = 0.07 with no statistical significance.
A. Thanks for the suggestions, changes were made to the text
"Table 2 presents the descriptive statistics for all variables, where critical thinking has the highest mean (31.97) with a standard deviation of 4.89. Additionally, the bivariate correlation matrix is shown, where study satisfaction is positively related to academic self-efficacy (r = 0.33, p < 0.01), critical thinking (r = 0.32, p < 0.01), and negatively with generalized anxiety (r = -0.14, p < 0.05). Likewise, academic self-efficacy is positively related to critical thinking (r = -0.36, p < 0.01) and negatively with generalized anxiety (r = -0.26, p < 0.01). Moreover, the relationship between critical thinking and generalized anxiety was not significant. "
- Table can be presented a bit better. What M and DE are, should be explained in the descriptive statistics.
A. Thanks for the suggestions, changes were made to the text
- Please specify what MLR stands for.
A. Thanks for the hints, it was specified that it is Multiple Linear Regression (MLR)

Round 2
Reviewer 1 Report
Dear Authors,
Thank you for your enhancing the research paper. However, I do recommend more enhancements could be done.
Best Regards,
Author Response
Thanks for the information, changes were made throughout the document

Reviewer 3 Report
After updates, Table 2 still has abbreviation DE without a full form. Legend has mentioned full form of M and SD, but not DE. Is it a typo?
Author Response
Thank you for recommending, we made the suggested changes
